# Differential Cytoophidium Assembly between *Saccharomyces cerevisiae* and *Schizosaccharomyces pombe*

**DOI:** 10.3390/ijms251810092

**Published:** 2024-09-19

**Authors:** Ruolan Deng, Yi-Lan Li, Ji-Long Liu

**Affiliations:** 1School of Life Science and Technology, ShanghaiTech University, Shanghai 201210, China; 2CAS Center for Excellence in Molecular Cell Science, Shanghai Institute of Biochemistry and Cell Biology, Chinese Academy of Sciences, Shanghai 200031, China; 3University of Chinese Academy of Sciences, Beijing 100049, China; 4Department of Physiology, Anatomy and Genetics, University of Oxford, Oxford OX1 3PT, UK

**Keywords:** co-culture, CTP synthase, Cts1, cytoophidium, *Saccharomyces cerevisiae*, *Schizosaccharomyces pombe*, Ura7

## Abstract

The de novo synthesis of cytidine 5′-triphosphate (CTP) is catalyzed by the enzyme CTP synthase (CTPS), which is known to form cytoophidia across all three domains of life. In this study, we use the budding yeast *Saccharomyces cerevisiae* and the fission yeast *Schizosaccharomyces pombe* as model organisms to compare cytoophidium assembly under external environmental and intracellular CTPS alterations. We observe that under low and high temperature conditions, cytoophidia in fission yeast gradually disassemble, while cytoophidia in budding yeast remain unaffected. The effect of pH changes on cytoophidia maintenance in the two yeast species is different. When cultured in the yeast-saturated cultured medium, cytoophidia in fission yeast disassemble, while cytoophidia in budding yeast gradually form. Overexpression of CTPS results in the presence and maintenance of cytoophidia in both yeast species from the log phase to the stationary phase. In summary, our results demonstrate differential cytoophidium assembly between *Saccharomyces cerevisiae* and *Schizosaccharomyces pombe*, the two most studied yeast species.

## 1. Introduction

Cytidine triphosphate (CTP) plays a crucial role in the synthesis of essential biomolecules within cells, including DNA, RNA, glycerophospholipids, and protein glycosylation [1,2]. CTP synthase (CTPS) is a key enzyme involved in CTP biosynthesis across various organisms. Notably, in 2010, studies revealed the presence of membraneless filamentous structures known as cytoophidia, formed by CTPS, in *Drosophila*, bacteria, and *S. cerevisiae* [3,4,5]. Subsequent research confirmed the existence of cytoophidia in human cells, *S. pombe*, *Arabidopsis thaliana*, and archaea [6,7,8,9,10,11], with recent discoveries also identifying cytoophidia in mouse thymus [12], highlighting the evolutionary conservation of these structures.

As the investigations advanced, the parallels and discrepancies in cytoophidia among diverse species became apparent. Research indicated that cytoophidia play a role in regulating cellular metabolism by adopting varying conformations to modulate CTPS enzymatic activity across organisms [13,14,15,16,17,18]. Treatment with the CTPS inhibitor 6-diazo-5-oxo-L-norleucine (DON) led to an increase in the number and length of cytoophidia in models such as *Caulobacter crescentus*, *Drosophila*, archaea, mammalian cells, fission yeast, and zebrafish [4,6,10,19,20]. The target of the rapamycin (TOR) pathway was found to influence cytoophidia formation in fission yeast, *Drosophila*, and mammalian cells [21,22], while cytoophidia were implicated in stress responses in *Drosophila* ovarian [23], yeast [16,24], and human cell lines [25]. Notably, cytoophidia were identified in rapidly proliferating cells, including *Drosophila* larvae neuroblasts [26], highly proliferative T cells in the murine thymus [12], and human cancer cells [7].

Comparing cytoophidia between budding and fission yeasts revealed both similarities and differences. Studies indicated that the expression products of the two CTPS genes, *URA7* and *URA8* in budding yeast [5,27], and in fission yeast, only one CTPS isoform encoded by *cts1,* could form cytoophidia [8]. In budding yeast, cytoophidia predominantly formed during the stationary phase, but not in the logarithmic phase [5,27]. In fission yeast, cytoophidia were only observed during the logarithmic phase but did not exist during the stationary phase [19,28]. Carbon deprivation induces the formation of cytoophidia in budding yeast, while the lack of a carbon source causes the depolymerization of cytoophidia in fission yeast [5,19]. In fission yeast, cytoophidia were sensitive to temperature changes, but not in budding yeast [5,19]. Despite these distinctions, studies highlighted shared features, such as the essential role of conserved histidine amino acids in CTPS polymerization [16,28] and the presence of cytoophidia in both the cytoplasm and nucleus of budding and fission yeasts [8,27]. These findings suggest distinct assembly and regulation mechanisms for cytoophidia in *S. pombe* and *S. cerevisiae*, as well as differing responses to environmental changes.

In this study, we utilized budding and fission yeasts as model organisms to compare cytoophidia dynamics under external environmental and intracellular CTPS alterations. Co-culturing fission yeast and budding yeast in YES+YPD medium, we observed that fission yeast cytoophidia gradually depolymerized under varying temperature conditions, while budding yeast cytoophidia remained unaffected. Furthermore, we found that pH variations and culture medium composition influenced cytoophidia maintenance differently in the two yeast species. Cytoophidia of fission yeast disappeared when cultured in the yeast-saturated cultured medium, but cytoophidia of budding yeast were gradually formed. The overexpression of CTPS led to the presence and maintenance of cytoophidia in both yeast species from the log phase to the stationary phase.

## 2. Results

### 2.1. Comparison of Cytoophidia between S. cerevisiae and S. pombe

The growth of yeast cells can be divided into four stages: the lag phase, logarithmic phase (log phase), stationary phase, and death phase. Each phase is characterized by unique metabolic conditions. However, the timing of cytoophidia appearance differs completely between fission yeast and budding yeast. We have engineered fission yeast with Cts1-YFP labeling and budding yeast with Ura7-mCherry labeling to investigate the presence of cytoophidia in both yeast species through cell culture observation. The results showed that cytoophidia were observed in over 90% of fission yeast cells during the log phase but disappeared during the stationary phase in fission yeast (Figure 1A,B,E). In contrast, cytoophidia were not present during the log phase but were formed in over 40% of budding yeast cells during the stationary phase (Figure 1C,D,F). This observation indicates that the formation and maintenance of cytoophidia in different yeast species were influenced by diverse conditions, suggesting that the regulatory mechanisms of cytoophidia formation and maintenance differed between *S. pombe* and *S. cerevisiae*. Therefore, we investigated the effects of various environmental changes on cytoophidia in budding yeast and fission yeast in the following content.

### 2.2. Co-Cultivation of Fission Yeast and Budding Yeast

To compare the influence of environment on the formation and maintenance of cytoophidia in fission yeast and budding yeast, we employed a co-cultivation approach, involving the simultaneous culture of both yeast species. Firstly, we selected three distinct culture media: fission yeast culture medium YES, budding yeast culture medium YPD, and a mixture medium of 50% YES and 50% YPD (YES+YPD). By co-culturing the two yeast types in the designated media and harvesting cells during the log and stationary phases (Figure 2A–C’), we evaluated the effects of YES, YPD, and YES+YPD media on both species. Our analysis revealed that the cell counts of fission yeast decreased when cultured in YPD compared to YES, while no significant difference was observed when cultured in YES+YPD (Figure 2D). Similarly, the cell counts of budding yeast decreased in YES but showed no significant difference when cultured in YES+YPD compared to YPD (Figure 2E).

Moreover, the proportion of cytoophidia-containing fission yeast remained consistent across the three media during co-cultivation (Figure 2F), but the proportion of cytoophidia-containing budding yeast decreased during culture in YES when compared with culture in YPD (Figure 2G). In YES, YPD, and YES+YPD, the period in which cytoophidia of both yeasts appeared was not affected. Based on these findings, we selected the YES+YPD medium for subsequent co-cultivation experiments.

### 2.3. Temperature Affects Cytoophidium Assembly in Fission Yeast but Not in Budding Yeast

To investigate the influence of high and low temperature conditions on the formation and maintenance of cytoophidia in yeast, we conducted co-cultivations of two yeast species at 37 °C and 4 °C, with cells co-cultured at 30 °C serving as the control. Through live-cell imaging, we observed that fission yeast cytoophidia gradually depolymerized under 37 °C during the log phase (Figure 3A–C). In contrast, budding yeast cytoophidia were not observed during the log phase but emerged in the stationary phase (Figure 3D).

Furthermore, the co-cultured yeasts during the log and stationary phases were transferred to 4 °C for 1 h, 2 h, and 3 h, respectively, before imaging analysis. We noted that compared to yeast co-cultured at 30 °C, fission yeast cytoophidia depolymerized under 4 °C during the log phase (Figure 3E–H and Figure 4). Interestingly, budding yeast cytoophidia remained unaffected at 4 °C (Figure 3I–L and Figure 4).

Analyzing data from fission yeast and budding yeast co-cultivated at different time points at both 4 °C and 37 °C, with cells co-cultured at 30 °C as the control, we observed a gradual decrease in the proportion of cytoophidia-containing fission yeast cells at both 4 °C and 37 °C compared to 30 °C (Figure 3M). In contrast, the proportion of cytoophidia-containing budding yeast cells remained unchanged at 4 °C and 37 °C (Figure 3N).

Moreover, measuring the OD600 values of fission yeast and budding yeast at 4 °C and 37 °C, with 30 °C as the control, revealed a deceleration in the cell growth rates of both yeast species at 37 °C (Figure 3O,P). Interestingly, the growth rates of both yeast species stagnated at 4 °C (Figure 4I,J). These results suggested that high and low temperature conditions impacted the maintenance of cytoophidia in fission yeast but did not affect the formation and maintenance of cytoophidia in budding yeast.

### 2.4. pH Values Influence Cytoophidium Assembly Differentially in Budding Yeast and Fission Yeast

To investigate the conditions affecting the formation and maintenance of cytoophidia, we first detected that the maintenance of cytoophidia of fission yeast was affected by the temperature of the culture environment. Then, we verified the influence of pH on cytoophidia in the culture environment. We co-cultured two yeast species in media with different pH values, as the control normal YES+YPD medium pH was 5.85. We observed that compared with the control group, fission yeast cytoophidia disappeared in the media with pH values of 7.0, 7.5, and 8.0, but polymerized at pH 4.0, 4.8, and 6.0 during the log phase (Figure 5A–G); cytoophidia were not formed in fission yeast during the stationary phase (Figure 5H–N). Budding yeast cytoophidia were not formed in the log phase (Figure 5A–G), but appeared in the stationary phase (Figure 5H–N).

We analyzed data from yeast co-cultured in media with different pH values, respectively. The results showed that compared with the control group, the proportion of cytoophidia-containing fission yeast was decreased in the media with pH 4.0, 6.0, 7.0, 7.5, and 8.0 in the log phase (Figure 5O). In budding yeast in the stationary phase, compared with the control group, the proportion of cytoophidia-containing cells decreased in media with pH 4.0, 7.5, and 8.0; but in medium with pH 7.0, the proportion of cytoophidia-containing cells increased (Figure 5P). Measurement via the OD600 value of fission yeast and budding yeast, respectively, in media with different pH values, showed that the cell growth rate of fission yeast was slowed down in media with pH 4.0, 6.0, 7.0, 7.5, and 8.0 (Figure 5Q). And the growth rate of budding yeast was slowed down in the media with pH 7.5 and 8.0 (Figure 5R). These data showed that both fission yeast and budding yeast cytoophidia were affected by pH in the culture environment.

### 2.5. Cytoophidia of Both Yeast Species Are Influenced by Yeast-Saturated Cultured Medium

The maintenance of fission yeast cytoophidia was affected by temperature and pH. We also investigated whether other cultural environments influenced the formation and maintenance of cytoophidia. We co-cultured two yeast species in yeast-saturated cultured medium, with the control yeast co-cultured in normal YES+YPD medium. The yeast co-cultured in the log phase was placed in the yeast-saturated cultured medium for 5 min, 15 min, 30 min, 1 h, 2 h, and 3 h, and then, yeasts were collected for imaging observation. We observed that compared to the control group, fission yeast cytoophidia were depolymerized gradually in the yeast-saturated cultured medium (Figure 6A–G). And budding yeast cytoophidia were gradually polymerized (Figure 6A–G).

We analyzed data from yeast co-cultured in yeast-saturated cultured medium during the log phase. The results showed that the proportion of cytoophidia-containing fission yeast gradually decreased compared with the control group, which was cultured in fresh medium (Figure 6H). The proportion of cytoophidia-containing budding yeast gradually increased compared with the control group (Figure 6I). Then, we measured the OD600 value of fission yeast and budding yeast cultured in the yeast-saturated cultured medium. Data showed that the cell growth rate of fission yeast and budding yeast stagnated (Figure 6J–K). These data showed that both fission yeast and budding yeast cytoophidia were affected when cultured in the yeast-saturated cultured medium and indicated that the growth and cytoophidia of these two yeast species were highly dependent on the nutritional environment.

### 2.6. Overexpression of CTPS Protein in Both Yeast Species Can Affect Cytoophidium Assembly

We conducted investigations into whether external environmental factors such as temperature, pH, and nutrient availability in the medium could impact the formation and maintenance of cytoophidia. Previous studies have indicated that changes in intracellular CTPS protein levels can influence cytoophidia in *Drosophila* [6,23,29,30,31]. Building on this, we explored whether the intracellular CTPS protein factor affected cytoophidia formation and maintenance in both fission yeast and budding yeast. We created strains of fission yeast and budding yeast that overexpressed CTPS proteins. Firstly, we overexpressed the Ura7 CTPS protein in budding yeast and observed the effects on cytoophidia formation. Our observations revealed that cytoophidia began to form in the log phase and persisted until the stationary phase, with the majority transitioning in shape from rod-like to round structures (Figure 7A–B”). 

Then, we overexpressed the Cts1 protein from fission yeast in budding yeast and observed the impact on cytoophidia formation. Similar to the previous experiment, cytoophidia formed during the log phase and persisted into the stationary phase, maintaining a rod-like shape characteristic of normal budding yeast (Figure 7C–D”).

The analysis of data from budding yeast strains overexpressing Ura7 or Cts1 revealed that, in comparison to the wild type in the stationary phase, the proportion of budding yeast cells containing cytoophidia decreased in both the log and stationary phases for Ura7 overexpression (Figure 7E). And the proportion of cytoophidia-containing budding yeast cells decreased in both the log and stationary phases for Cts1 overexpression compared to the wild type in the stationary phase (Figure 7F).

Subsequently, we overexpressed the CTPS protein Cts1 in fission yeast, resulting in an exceedingly low number of cytoophidia-containing cells, approximately 1.5%, being detected in both the log and stationary phases (Figure 8E). Notably, the shape of cytoophidia during the stationary phase appeared elongated compared to the wild type (Figure 8A–B”). Furthermore, when we introduced the CTPS protein Ura7 from budding yeast into fission yeast, cytoophidia formation initiated in the log phase and persisted until the stationary phase, with the cytoophidia exhibiting increased length compared to the wild type (Figure 8C–D”). Additionally, the proportion of cytoophidia-containing cells overexpressing Ura7 decreased in both the log and stationary phases compared to the wild type in the log phase (Figure 8F).

We also measured the OD600 value of fission yeast and budding yeast overexpressing CTPS proteins, which indicated a deceleration in the growth rates of both yeast species (Figure 7G and Figure 8G). These findings highlight that the formation and maintenance of cytoophidia in fission yeast and budding yeast overexpressing CTPS proteins were notably impacted.

### 2.7. Interspecies Overexpression of Ura7 Generates Giant Cytoophidia crossing Cell Membrane

In fission yeast overexpressing Ura7, we observed a peculiar phenomenon where the cytoophidia elongated and extended across two cells. Through live-cell imaging, we tracked this event and discovered that the giant cytoophidia spanning both cells took a considerable amount of time to dissociate (Figure 9A). To further investigate this phenomenon, we utilized Calcofluor White Staining to stain and observe the cells, with red representing the cytoplasm, blue indicating the cell wall, and the cytoophidia being marked in green by endogenous YFP. Our observations revealed that the cytoophidia indeed traversed the membrane and cell wall of both cells following cytoplasmic division (Figure 9B–K). These findings suggest that the overexpression of Ura7 induced cytoophidia elongation, with some cytoophidia capable of crossing the cell membrane and existing in two cells during cell division (Figure 9L).

## 3. Discussion

Although cytoophidia have been extensively studied and are common in various species, differences in cytoophidia exist among different species, including disparities between *S. cerevisiae* and *S. pombe*. However, there have been few reports systematically comparing the differences in cytoophidia between these species under varying conditions. In this study, we used these two types of yeasts as model organisms to compare the effects of the external environment and intracellular CTPS protein on cytoophidia.

### 3.1. Comparison of Cytoophidia in S. cerevisiae and S. pombe under External Environmental Alterations

In this study, we utilized a co-cultivation approach to grow two yeast species in the same system, allowing for a comparative analysis of how cytoophidia responded to identical environmental conditions in both species. We explored the effects of external factors, such as extreme temperatures, pH variations, and nutrient deficiencies, on cytoophidia in both yeast species. Our findings revealed that cytoophidia in fission yeast were highly sensitive to environmental changes, particularly to temperature fluctuations, pH shifts, and nutrient scarcity, resulting in a decrease in the proportion of cells containing cytoophidia or even the disassembly of cytoophidia. These results were consistent with previous studies [19,32], suggesting that maintaining and regulating cytoophidia in fission yeast requires a relatively stable living environment.

Regarding budding yeast, we showed that cytoophidia exhibited greater tolerance to temperature variations, showing less susceptibility to high or low temperatures, consistent with previous reports [5]. However, pH alterations did impact cytoophidia in budding yeast. Previous studies indicated that a decrease in intracellular pH from 7.0 to 6.0 through cell membrane permeation increased the proportion of cells containing cytoophidia [16]. In our research, we observed that at extracellular pH levels of 4.0, 7.5, and 8.0, the proportion of cytoophidia-containing cells decreased, while at pH 7.0, the proportion increased. These discrepancies from previous findings may be attributed to differences between extracellular and intracellular pH levels, as we did not permeate the cell membrane.

Prior studies revealed that carbon deprivation triggers cytoophidia assembly in budding yeast [5]. Our data provided further insights into this process, showing that cultivating yeast in a yeast-saturated cultured medium induced cytoophidia formation within approximately 5 min, with the proportion of cytoophidia-containing cells increasing over time. This phenomenon was akin to the observations in budding yeast when cultured to the stationary phase, suggesting the presence of factors in the yeast-saturated cultured medium that induce cytoophidia formation in budding yeast. Furthermore, this study represents the first systematic comparison of cytoophidia in *S. cerevisiae* and *S. pombe* under external environmental alterations.

### 3.2. Comparison of Cytoophidia in S. cerevisiae and S. pombe under Intracellular CTPS Alterations

In previous studies, it was observed that the morphology and abundance of cytoophidia were influenced by the CTPS protein within the cell. For instance, in *Drosophila* follicle cells, the downregulation of CTPS using RNAi technology led to cytoophidia disassembly, while the overexpression of CTPS promoted cytoophidia formation [6,29]. In *Drosophila* ovaries, cytoophidia elongated under CTPS overexpression [23]. Additionally, in *Drosophila* adipose tissue and posterior follicle cells, CTPS overexpression increased cytoophidia abundance and prolonged cytoophidia length [30,31]. These observations suggest that the expression levels of the CTP synthase protein are crucial for maintaining the integrity of cytoophidia. Structural analyses of CTP synthase filaments have revealed that in budding yeast, the assembly of CTPS cytoophidia inhibits enzyme activity. When cells require increased CTPS activity, CTPS molecules on cytoophidia are released into the cytoplasm to elevate the concentration of free CTPS and enhance the synthesis reaction rate [16]. Cytoophidia are formed during periods of non-cell proliferation in budding yeast. The structure of cytoophidia in fission yeast has not been elucidated yet, but we speculate that the formation of cytoophidia in fission yeast may enhance enzyme activity, as these cytoophidia form during periods of rapid cell proliferation.

In our study, in budding yeast, overexpression of its own CTPS protein (Ura7) promoted the formation of cytoophidia starting in the logarithmic phase and persisting into the stationary phase, with a change in shape from elongated to punctate. This phenomenon is similar to the overexpression of CTPS in *Drosophila* embryos, both of which enhance cytoophidia formation. The overexpression of CTPS in *Drosophila* embryos promotes cytoophidia formation from stage 1 of embryonic development. However, a notable difference is that while the overexpression of CTPS in budding yeast leads to a slowdown in cell growth rate, the overexpression of CTPS in *Drosophila* embryos does not cause any defects in embryo development. In fission yeast, overexpression of its own protein (Cts1) almost completely eliminates the presence of cytoophidia, with only extremely rare instances of elongated cytoophidia observed in a few cells. This is in stark contrast to the overexpression of CTPS in budding yeast, and this phenomenon has not been observed in other species so far.

We speculate that in budding yeast, the decrease in cell growth rate after overexpression of Ura7 is due to the excessively high concentration of Ura7 in the cells. This high concentration leads to the premature assembly of cytoophidia, which subsequently inhibits enzyme activity to ensure cell survival. On the other hand, in fission yeast, the slowing of cell growth rate after overexpression of Cts1 is attributed to the excessively high concentration of Cts1 in the cells. In this scenario, cytoophidia assembly is unnecessary to enhance enzyme activity for cell survival, resulting in cytoophidia disassembly.

In addition, we observed that heterologous overexpression of Cts1 in budding yeast and Ura7 in fission yeast resulted in the co-localization of Ura7 and Cts1 on cytoophidia in both yeast species. These cytoophidia formed during the logarithmic phase and persisted into the stationary phase in both yeasts. We speculate that Ura7 may influence Cts1 disassembly on cytoophidia in fission yeast, leading to the presence of cytoophidia in the stationary phase. Similarly, Cts1 may impact Ura7 assembly in budding yeast, resulting in cytoophidia formation as early as the logarithmic phase. We hypothesize that the variances in structure and function of cytoophidia formed by Cts1 and Ura7 in these two yeast species lead to alterations in the shape of the cytoophidia compared to the wild type, ultimately resulting in a slower yeast growth rate. We posit that variations exist in the structure, regulatory mechanisms, and functions of the cytoophidia formed by Ura7 and Cts1 in budding yeast and fission yeast. Further investigation of these distinctions is imperative to deepen our comprehension.

In conclusion, the overexpression of CTPS proteins (Ura7 or Cts1) in both yeast species altered the timing of cytoophidia formation, the proportion of cells containing cytoophidia, and the morphology of cytoophidia. These findings suggest that intracellular CTPS protein could directly affect the formation and maintenance of cytoophidia, despite differences in the assembly and regulation mechanisms of cytoophidia between the two yeast species and their responses to environmental changes. This study represents the first investigation comparing cytoophidia under intracellular CTPS alterations in budding yeast and fission yeast.

### 3.3. Connection between Cytoophidia and Cell Growth

In previous studies, it was consistently demonstrated that cytoophidia were closely linked to cell growth rates. In both fission yeast and budding yeast, a conserved histidine mutation in the CTPS, changing histidine to alanine, resulted in the absence of cytoophidia formation, leading to a slower cell growth rate [16,28]. Additionally, in fission yeast, both the knockdown and overexpression of CTPS were found to impact cytoophidia formation and decrease cell growth rates [28]. In *Drosophila* posterior follicle cells, the knockdown of CTPS was shown to alleviate the over-proliferative phenotype in *hpo* mutant cells [30]. Furthermore, in human cancer cells like MKN45, the overexpression of CTPS was found to promote cytoophidia formation and decelerate cell proliferation rates [33].

In our study, we observed that the growth rates of both yeast species were influenced by changes in extracellular environmental factors such as temperature, pH, nutrient availability in the medium, and the intracellular overexpression of CTPS. Cell growth in both yeast types was notably hindered in low temperatures and when cultured in yeast-saturated medium. Moreover, the presence of cytoophidia in both fission yeast and budding yeast was affected by these external and internal factors. These findings align with previous research and suggest a correlation between cytoophidia and cell growth. We hypothesize that there may be a reciprocal regulation between cytoophidia and cell proliferation, although the specific regulatory mechanism remains unknown and warrants further investigation.

## 4. Materials and Methods

### 4.1. Yeast Strain and Culture Medium

The budding yeast strain was haploid and derived from the wild-type haploid (BY4741) as previously described [24]. The budding yeast strain with endogenous mCherry tagged Ura7 was created by homologous recombination. The fission yeast strain with endogenous YFP tagged Cts1 was constructed as previously described [28]. Budding yeast was cultured in complete medium (YPD) or basic medium with necessary amino acids at 30 °C. Fission yeast was cultured in complete medium (YES) or Edinburgh minimal medium at 30 °C as previously described [28]. For plate culture, an additional 2% agar was added and incubated at a temperature of 30 °C for 3–5 days.

### 4.2. Plasmid Construction

The overexpression plasmids were constructed using the ClonExpress Ultra One Step Cloning kit (Vazyme, Nanjing, China. Cat. C115-01); genes for the overexpressed plasmids were amplified by PCR from the yeast genome and integrated into corresponding vectors with specific nutritional deficiencies. To construct a plasmid for overexpressing *cts1* and *URA7* genes, fission yeast genomic DNA was used as a template for PCR amplification to obtain the *cts1* gene, and budding yeast genomic DNA was used as a template for PCR amplification to obtain the *URA7* gene, then inserted gene into vectors.

For the plasmids overexpressed in budding yeast, we constructed a relatively stable episomal plasmid using an *ARS/CEN* (*S. cerevisiae* CEN6 centromere fused to an autonomously replicating sequence) replication origin. The Ura7-mGFP or Cts1-miRFP670nano coding genes were inserted into this plasmid, which contained the His3 (Imidazoleglycerol-phosphate dehydratase) auxotrophic selection marker.

For the plasmids overexpressed in fission yeast, we constructed a relatively stable episomal plasmid using an *ars1* (*S. pombe* autonomously replicating sequence *ars1*) replication origin. The Cts1-miRFP670nano or Ura7-miRFP670nano coding genes were inserted into this plasmid, which contained the Ade2 (phosphoribosylaminoimidazole carboxylase) auxotrophic selection marker.

All constructed plasmids underwent verification of their DNA sequences by Sanger sequencing. Plasmids were transferred into yeast using the lithium acetate method [34].

### 4.3. Growth Assays

Cells were cultured in a complete medium in at least two passages to maintain high proliferative activity. The growth curve was plotted by measuring OD600 after reviving cells to a robust growth state and diluting them to an initial OD600 of ~0.05 or lower concentrations. Optical density was measured at corresponding time points during culture. All cells were cultured at 30 °C.

### 4.4. Cell Fixation, Image Acquisition, and Live-Cell Imaging

For fixed samples, yeast cells were collected and fixed in 4% paraformaldehyde for 10 min at 30 °C with shaking at 250 rpm. The fixed cells were collected and washed with 1 × PBS at once and resuspended in 1 × PBS. The yeast suspension and 1.5% low-melting agarose were mixed at a ratio of 2:1 before being dropped onto a glass slide. After covering the sample with a coverslip, the film was sealed using nail polish. Images of the fixed samples were acquired using a Zeiss LSM 980 Airyscan2 microscope (Zeiss, Oberkochen, Germany) equipped with a Plan APO 63 ×/1.40 OIL objective (Zeiss, Oberkochen, Germany) in Airyscan mode.

Live-cell imaging was carried out by referring to our laboratory’s previous protocol [35]; briefly, yeast cells were cultured in glass-bottomed Petri dishes (Thermo scientific^TM^, Waltham, MA, USA. Cat. 150460). Yeast was cultured until it reached an exponential state. Then, approximately 0.5 μL of yeast culture was placed at the center of the glass-bottomed Petri dish, followed by a gentle addition of dropwise 1.5% low-melting agarose solution onto the yeast. After allowing it to stand at room temperature for 10 min, the culture medium (about 1 mL) was added. A Zeiss Cell Observer SD spinning disk confocal microscope (Zeiss, Oberkochen, Germany) equipped with a 63× OIL objective was used for live-cell imaging at 30 °C.

### 4.5. Quantification and Statistical Analysis

The sample size for each figure is indicated in the figure legends. Statistical significance between conditions was assessed using unpaired Student’s *t*-test. For multiple group comparisons, one-way ANOVA analysis was performed. All statistical analyses were performed in GraphPad Prism, all error bars represent the mean ± standard errors (S.E.M.), and significance is denoted as * *p* < 0.05, ** *p* < 0.01, *** *p* < 0.001, and **** *p* < 0.0001; ns denotes not significant.

## Figures and Tables

**Figure 1 ijms-25-10092-f001:**
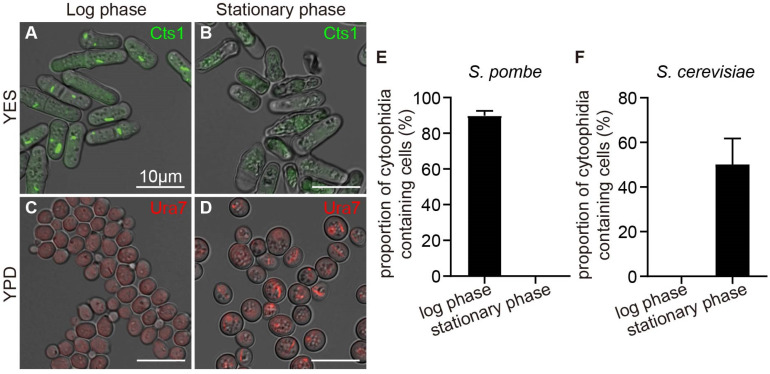
Cytoophidia appear at different phases in two yeast species. (**A**,**B**) Cytoophidia formed during log phase (**A**) but disappeared in stationary phase (**B**) in fission yeast cultured in YES. (**C**,**D**) Cytoophidia were absent in log phase (**C**) but formed during stationary phase (**D**) in budding yeast cultured in YPD. Channels: green (Cts1), red (Ura7), and bright field. Scale bars, 10 μm. (**E**) Proportion of cytoophidia-containing cells in (**A**,**B**) (>440 cells were manually measured). (**F**) Proportion of cytoophidia-containing cells in (**C**,**D**) (>636 cells were manually measured). All values are mean ± S.E.M.

**Figure 2 ijms-25-10092-f002:**
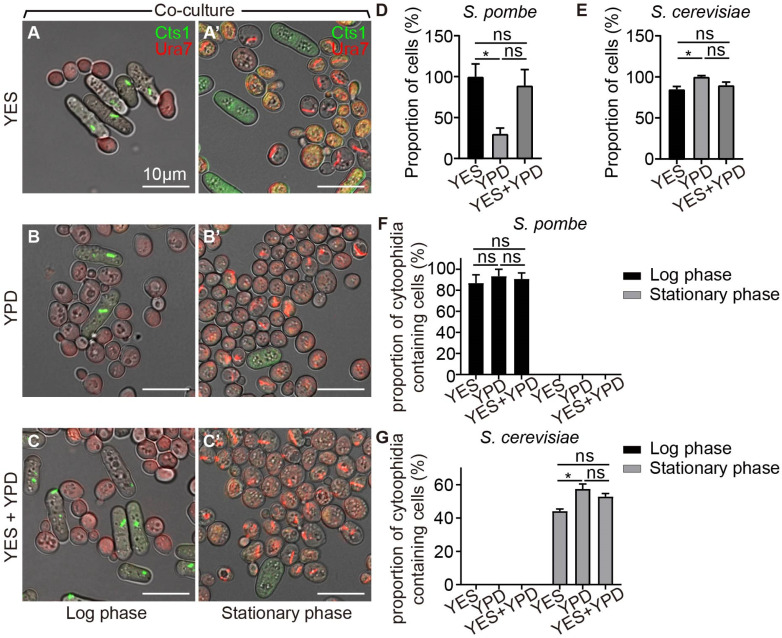
Co-culture of fission yeast and budding yeast. (**A**–**C’**) Fission yeast and budding yeast were co-cultured during the log phase and stationary phase in YES (**A**,**A’**), YPD (**B**,**B’**), and 50% YES+50% YPD mixture (**C**,**C’**). Channels: green (Cts1), red (Ura7), and bright field. Scale bars, 10 μm. (**D**) Proportion of fission yeast in co-culture during stationary phase in YES, YPD, and YES+YPD media (>328 cells were manually measured). (**E**) Proportion of budding yeast in co-culture in stationary phase in YES, YPD, and YES+YPD medium (>2147 cells were manually measured). (**F**) Percentage of cells containing cytoophidia in fission yeast in YES, YPD, and YES+YPD medium (>398 cells were manually measured). (**G**) Percentage of cells containing cytoophidia in budding yeast in YES, YPD, and YES+YPD medium (>3440 cells were manually measured). All values are mean ± S.E.M. *, *p* < 0.05; ns, no significant difference.

**Figure 3 ijms-25-10092-f003:**
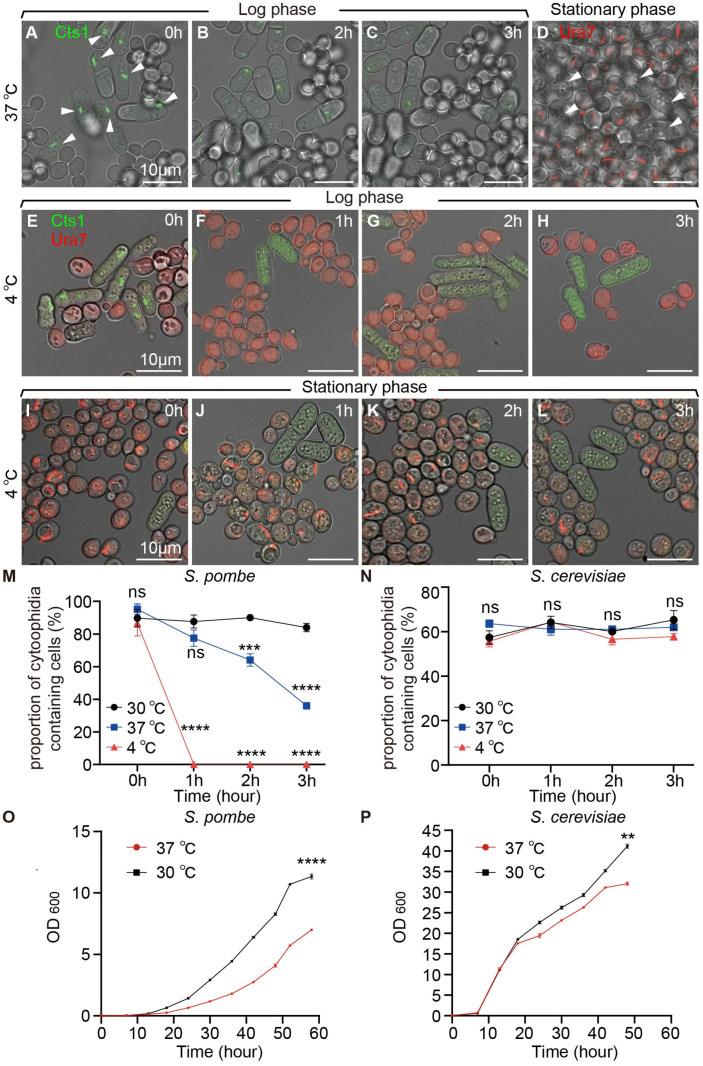
Cytoophidia of fission yeast were affected under high and low temperature conditions. (**A**–**C**) Live-cell imaging photos of co-cultured fission yeast and budding yeast in YES+YPD medium; white arrows showed that as culture time increased at 37 °C, cytoophidia gradually disappeared in fission yeast. (**D**) Cytoophidia formed in the stationary phase when budding yeast was cultured in 37 °C conditions. White arrows showed the absence of cytoophidia in fission yeast during the stationary phase. (**E**–**H**) Yeast in log phase cocultured at 4 °C for 0 h (**E**), 1 h (**F**), 2 h (**G**), and 3 h (**H**). (**I**–**L**) Yeast in stationary phase cocultured in 4 °C for 0 h (**I**), 1 h (**J**), 2 h (**K**), and 3 h (**L**). Channels: green (Cts1), red (Ura7), and bright field. Scale bars, 10 μm. (**M**) Proportion of cytoophidia-containing cells of fission yeast at log phase at 37 °C, 4 °C, and 30 °C (>1700 cells were manually measured). (**N**) Proportion of cytoophidia-containing cells of budding yeast in stationary phase at 37 °C, 4 °C, and 30 °C (>5000 cells were manually measured). (**O**) Growth curves of fission yeast at 30 °C and 37 °C. (*n* = 2). (**P**) Growth curves of budding yeast at 30 °C and 37 °C. (*n* = 2). All values are mean ± S.E.M. ****, *p* < 0.0001; ***, *p* < 0.001; **, *p* < 0.01; ns, no significant difference.

**Figure 4 ijms-25-10092-f004:**
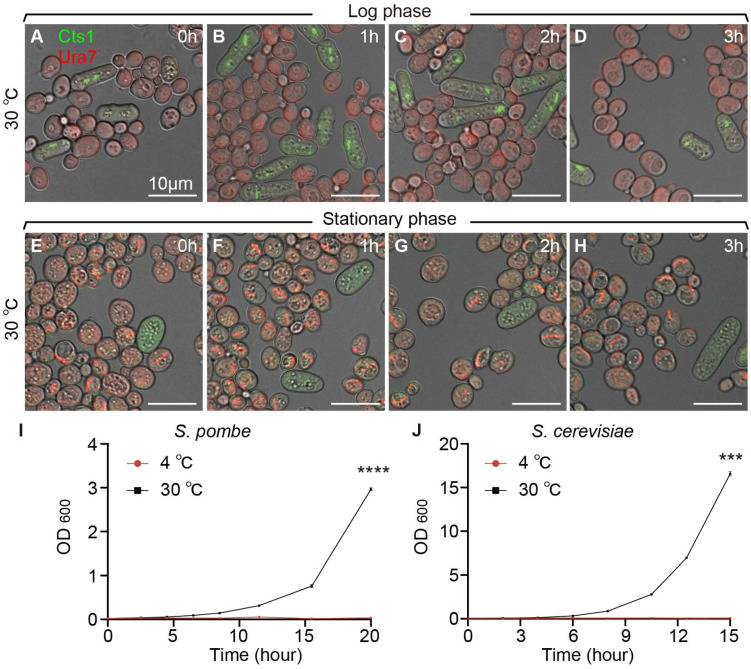
Fission yeast and budding yeast coculture under 30 °C conditions. (**A**–**D**) Yeasts in log phase cocultured at 30 °C for 0 h (**A**), 1 h (**B**), 2 h (**C**), and 3 h (**D**). (**E**–**H**) Yeasts in stationary phase cocultured at 30 °C for 0 h (**E**), 1 h (**F**), 2 h (**G**), and 3 h (**H**). Channels: green (Cts1), red (Ura7), and bright field. Scale bars, 10 μm. (**I**) Growth curves of fission yeast at 30 °C and 4 °C. (*n* = 2). (**J**) Growth curves of budding yeast at 30 °C and 4 °C. (*n* = 2). All values are mean ± S.E.M. ****, *p* < 0.0001; ***, *p* < 0.001.

**Figure 5 ijms-25-10092-f005:**
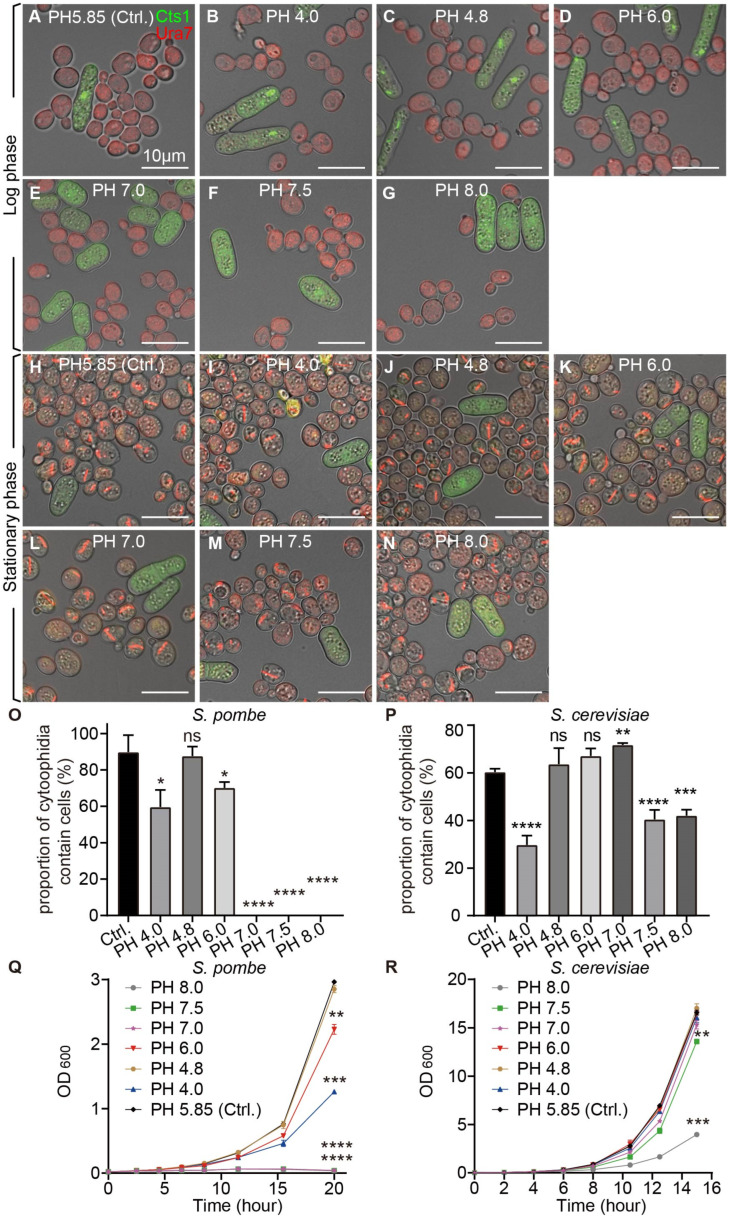
Cytoophidia of both yeast species were influenced by pH conditions. (**A**–**G**) Photos of yeast in log phase cocultured in the medium of pH 5.85, as control (**A**), in the medium of pH 4.0 (**B**), in the medium of pH 4.8 (**C**), in the medium of pH 6.0 (**D**), in the medium of pH 7.0 (**E**), in the medium of pH 7.5 (**F**), and in the medium of pH 8.0 (**G**). (**H**–**N**) Photos of yeast in stationary phase cocultured in the medium of pH 5.85, as control (**H**), in the medium of pH 4.0 (**I**), in the medium of pH 4.8 (**J**), in the medium of pH 6.0 (**K**), in the medium of pH 7.0 (**L**), in the medium of pH 7.5 (**M**), and in the medium of pH 8.0 (**N**). Channels: green (Cts1), red (Ura7), and bright field. Scale bars, 10 μm. (**O**) Proportion of cytoophidia-containing cells of fission yeast in log phase at pH 5.85 (Ctrl.), pH 4.0, pH 4.8, pH 6.0, pH 7.0, pH 7.5, and pH 8.0. (>615 cells were manually measured). (**P**) Proportion of cytoophidia-containing cells of budding yeast in stationary phase at pH 5.85 (Ctrl.), pH 4.0, pH 4.8, pH 6.0, pH 7.0, pH 7.5, and pH 8.0. (>1000 cells were manually measured). (**Q**) Growth curves of fission yeast at PH 5.85 (Ctrl.), PH 4.0, PH 4.8, PH 6.0, PH 7.0, PH 7.5, and PH 8.0. (*n* = 2). (**R**) Growth curves of budding yeast at PH 5.85 (Ctrl.), PH 4.0, PH 4.8, PH 6.0, PH 7.0, PH 7.5, and PH 8.0. (*n* = 2). All values are mean ± S.E.M. ****, *p* < 0.0001; ***, *p* < 0.001; **, *p* < 0.01, *, *p* < 0.05; ns, no significant difference.

**Figure 6 ijms-25-10092-f006:**
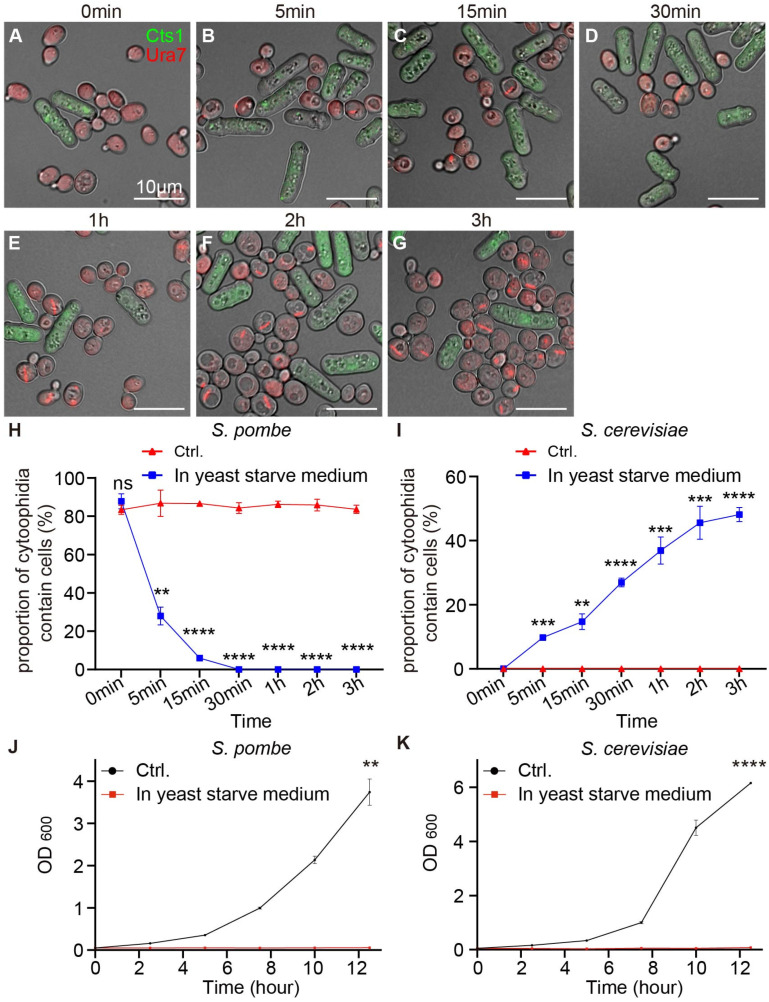
Cytoophidia of both yeast species were influenced by yeast-saturated cultured medium. (**A**–**G**) Yeast in log phase co-cultured in the yeast-saturated cultured medium for 0 min (**A**), 5 min (**B**), 15 min (**C**), 30 min (**D**), 1 h (**E**), 2 h (**F**), and 3 h (**G**). Channels: green (Cts1), red (Ura7), and bright field. Scale bars, 10 μm. (**H**) Proportion of fission yeast cells with cytoophidia during the log phase in the yeast-saturated cultured medium (>300 cells were manually measured). (**I**) Proportion of budding yeast cells with cytoophidia during different periods in log phase in the yeast-saturated cultured medium (>1000 cells were manually measured). (**J**) Growth curves of fission yeast cultured in yeast-saturated cultured medium. (*n* = 2). (**K**) Growth curves of budding yeast cultured in yeast-saturated cultured medium. (*n* = 2). All values are mean ± S.E.M. ****, *p* < 0.0001; ***, *p* < 0.001; **, *p* < 0.01; ns, no significant difference.

**Figure 7 ijms-25-10092-f007:**
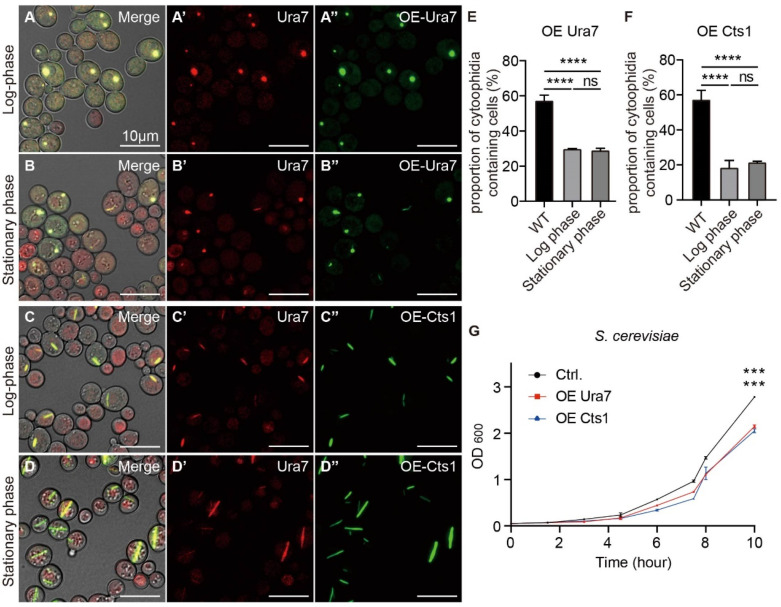
Cytoophidia were affected by the overexpression of CTPS protein in *S. cerevisiae*. (**A**–**B”**) Overexpression of Ura7 in budding yeast resulted in the formation of cytoophidia, as dot-like structures, in the log phase and stationary phase. Channels: red (endogenous Ura7-mCherry), green (overexpressed Ura7-mGFP), and bright field. Scale bars, 10 μm. (**C**–**D”**) Overexpression of Cts1 in budding yeast resulted in the formation of cytoophidia, as rod-like structures, in the log phase and stationary phase. Channels: red (endogenous Ura7-mCherry), green (overexpressed Cts1-miRFP670nano), and bright field. Scale bars, 10 μm. (**E**) Proportion of cytoophidia-containing cells of budding yeast overexpressing Ura7 (>1580 cells were manually measured). (**F**) Proportion of cytoophidia-containing cells of budding yeast overexpressing Cts1 (>1700 cells were manually measured). (**G**) Growth curves of budding yeast overexpressing CTPS protein. (*n* = 2). All values are mean ± S.E.M. ****, *p* < 0.0001; ***, *p* < 0.001; ns, no significant difference.

**Figure 8 ijms-25-10092-f008:**
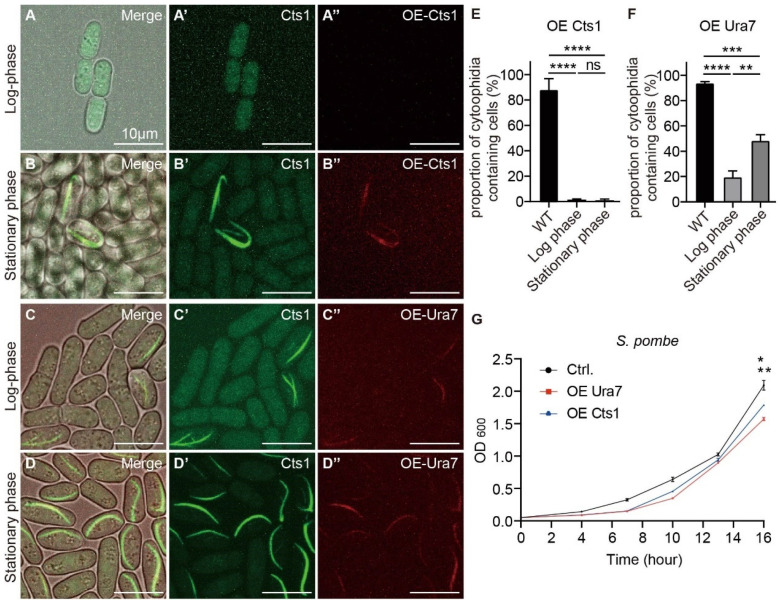
Cytoophidia were affected by the overexpression of CTPS in *S. pombe*. (**A**–**B”**) Overexpression of Cts1in fission yeast resulted in the rare occurrence of cytoophidia in the log phase and stationary phase, and the shape of cytoophidia appeared elongated. Channels: green (endogenous Cts1-YFP), red (overexpressed Cts1-miRFP670nano), and bright field. Scale bars, 10 μm. (**C**–**D”**) Overexpression of Ura7 in fission yeast led to the development of elongated cytoophidia in both the log phase and stationary phase. Channels: green (endogenous Cts1-YFP), red (overexpressed Ura7-miRFP670nano), and bright field. Scale bars, 10 μm. (**E**) Proportion of cytoophidia-containing cells of fission yeast overexpressing Cts1 (>660 cells were manually measured). (**F**) Proportion of cytoophidia-containing cells of fission yeast overexpressing Ura7 (>1500 cells were manually measured). (**G**) Growth curves of fission yeast overexpressing CTPS protein. (*n* = 2). All values are mean ± S.E.M. ****, *p* < 0.0001; ***, *p* < 0.001; **, *p* < 0.01; *, *p* < 0.05; ns, no significant difference.

**Figure 9 ijms-25-10092-f009:**
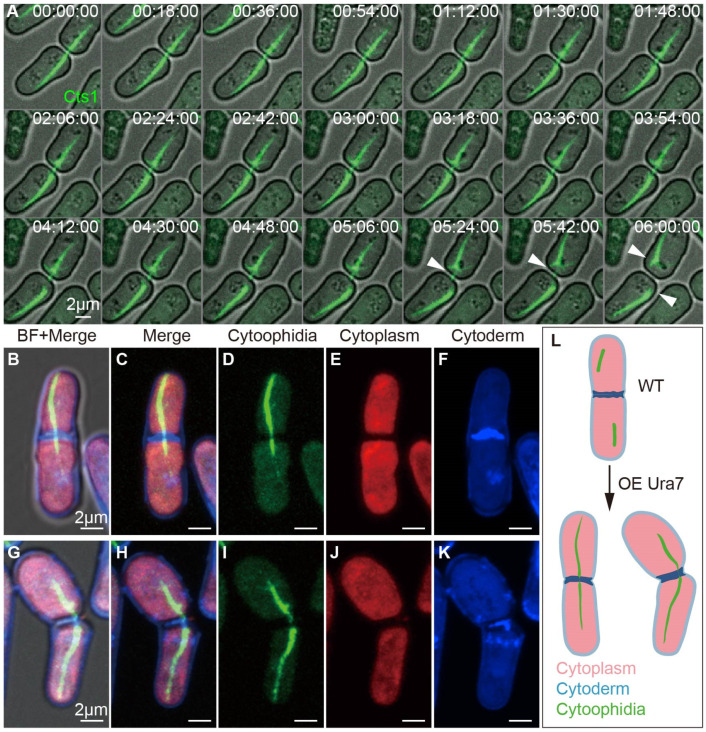
Cytoophidia of fission yeast overexpressing Ura7 became longer and crossed the cell membrane. (**A**) Snapshots of 6 h of live-cell imaging, recording the process of cytoophidium rupture across two cells. CTP synthase (CTPS) fused to YFP was expressed under the endogenous promoter and Ura7 was overexpressed in fission yeast. The break of the cytoophidium across both cells under prolonged pulling, as shown by the white arrow. Channels: green (Cts1) and bright field. Scale bars, 2 μm. (**B**–**F**) Photograph of fission yeast overexpressing Ura7-miRFP670nano; cytoophidium traversed two fission yeasts that had undergone cytoplasmic division. (**G**–**K**) Photograph of fission yeast overexpressing Ura7-miRFP670nano; cytoophidium passed through two fission yeasts that had undergone cytoplasmic division and were truncated. Channels: green (Cts1), red (cytoplasm), blue (cytoderm), and bright field. Scale bars, 2 μm. (**L**) Schematic diagram of cytoophidia in fission yeast overexpressing Ura7.

## Data Availability

Data is contained within the article.

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
