# Peer review of "Differential Cytoophidium Assembly between Saccharomyces cerevisiae and Schizosaccharomyces pombe"

_ijms, 2024, doi:10.3390/ijms251810092_

Round 1
Reviewer 1 Report
Comments and Suggestions for Authors
Deng et al. present a research manuscript describing interesting work in systematically comparing cytoophiium dynamics in two widely-used yeast organisms, S. cerevisiae and S. pombe. By endogenously tagging proteins that are known to be components of cytoophiiums and following the structure formations using live microscopy, the authors test different growth and environmental conditions which are commonly looked at by yeast researchers. Overall, this work will positively add to the existing body of cytoophiium knowledge. Below are a few points that the authors should consider in their manuscript revision:
1. There are two instances of grammatical errors (“not exist” on line 53 and “temperature sensitive” on line 56). Overall, the English writing was very sound.
2. It is unclear from the text (Results and Material and Methods sections) whether the overexpressed Ura4 and Cts1 proteins were also tagged. If they were, that information should also be clarified.
3. The overexpression experiments yielded intriguing results. The Discussion section mainly reiterates the overexpression experimental results but does not discuss how those effects could be mediated or potentially what the biological significance could be. It would be helpful for the authors to expand that part of the Discussion.
Comments on the Quality of English LanguageVery few edits appear required.
Reviewer 2 Report
Comments and Suggestions for Authors
Cytidine triphosphate plays a crucial role in the synthesis of essential biomolecules within cells, CTP synthase is a key enzyme involved in CTP biosynthesis across various organisms. Studies have revealed the presence of membraneless cytoophidia, formed by CTP synthase. Cytoophidia play essential role in regulating cellular metabolism by modulate CTP synthase enzymatic activity. Authors performed investigations into external environmental factors such as temperature, pH, nutrient and force expression to see whether thses pressure could impact the formation and maintenance of cytoophidia. Authors have shown several interesting findings, and the manuscript is well writen. I have some question, and hope authors to answer them and make minior change of the manuscript.
1. The proportion of budding yeast seems changed when cultured in the YES+YPD medium? (Fig 2c, c’)
2. The figures in Fig 3 and fig 4, if look at the growth curve, is not corresponding to above figures/span/p p style="margin-left: 36pt; text-indent: 0pt"3. In Fig 5, F, G, is better to select the image with two or more Cts1 marked cells
4. For the overexpression, it is better increasing the target gene expression with different amount vector, to see whether has similar effect, as in yeast is different in Drosophila, and authors shall discuss the difference.
5. It would be great to check whether overexpression Cts1 and Ura7 have similar result after depletion endogenous Cts1 and Ura7.
Comments on the Quality of English LanguageThe written is OK
